# Location Problem in Relativistic Positioning: Relative Formulation

**Ramón Serrano Montesinos** [1,*] , **Joan Josep Ferrando** [1,2] and **Juan Antonio Morales-Lladosa** [1,2]

1    Departament d'Astronomia i Astrofísica, Universitat de València, 46100 Burjassot, Spain;
     joan.ferrando@uv.es (J.J.F.); antonio.morales@uv.es (J.A.M.-L.)
2    Observatori Astronòmic, Universitat de València, 46980 Paterna, Spain
*    Correspondence: rasemon@alumni.uv.es

**Abstract:** A relativistic positioning system is a set of four emitters broadcasting their proper times by means of light signals. The four emitter times received at an event constitute the emission coordinates of the event. The covariant quantities associated with relativistic positioning systems are analysed relative to an observer in Minkowski space-time by splitting them in their relative space-like and time-like components. The location of a user in inertial coordinates from a standard set of emission data (emitted times and satellite trajectories) is solved in the underlying 3+1 formalism. The analytical location solution obtained by Kleusberg for the GPS system is recovered and interpreted in a Minkowskian context.

**Keywords:** relativistic positioning systems; pseudorange navigation equations; Kleusberg's solution

## 1. Introduction

The central question in positioning theories is to determine the location of the user without ambiguity after solving the navigation equations.

For users of global navigation satellite systems (GNSSs), such as GPS and Galileo, these navigation equations are usually formulated in terms of pseudoranges, which are the apparent distances to the user from each of the emitters, as inferred from the travel time of the signal (see Section 4 for more detail).

In general, the procedures used in GNSSs to solve the equations analytically can be divided into two classes depending on whether they use pseudoranges (Bancroft's algorithm [1]) or pseudorange differences (Kleusberg's method [2,3]), thus eliminating the user clock bias (see Section 4). In line with [1], Abel and Chaffee stated the problem using the Lorentz scalar product [4] and analysed the existence and non-unicity of the solutions [5] (bifurcation), which was also considered in [6].

A fully relativistic formulation of the problem in Minkowski space-time was given in the context of the theory of relativistic positioning systems (RPSs) [7,8]. For the foundations, genesis, objectives and perspectives of the RPS theory, refer to [9–12] and references therein. Recently, Bancroft's solution [1] was interpreted in the language of RPSs [13,14], but the corresponding RPS interpretation of Kleusberg's solution remained to be done. This complementary task is achieved in this paper.

Recall that an RPS is a set of four ordered clocks $A$ ($A = 1, 2, 3, 4$) with world-lines $\gamma_A(\tau^A)$ broadcasting their times $\tau^A$ by means of light signals. For simplicity, and as is the case in the present work, the time considered is the proper time. The four times $\{\tau^A\}$ received by an event $x$ constitute the *emission coordinates* of the event [15]. Again, for simplicity, the emission and reception processes are assumed to be continuous. For the theory and classification of RPSs based on a discrete set of data, see [16].

Current GNSSs do not broadcast the proper time of their clocks, but the system's own time (the GPS or the Galileo time), a time which, roughly speaking, coincides up to a fixed shift with the International Atomic Time.

In any case, for every set of four satellites in a given constellation (that could include both GPS and Galileo satellites), the four broadcast times essentially share the algebraic and differential properties that characterise an emission coordinate system. These properties were analysed elsewhere [7,15], and they do not need to be exhaustively remembered here. Only one property needs to be highlighted now, which concerns the very unusual character of the emission coordinates: all the gradients $d\tau^A$ are light-like. This means that $\{\tau^A\}$ is a set of four null gradient coordinates, which is an outstanding property that wholly determines the causal class of every relativistic emission coordinate system [17].

Suppose there is a specific coordinate system $\{x^\alpha\}$ that covers the whole region of emission coordinates, let $\gamma_A(\tau^A)$ be the world-lines of the clocks $A$ with respect to this particular coordinate system and let $\{\tau^A\}$ be the values of the emission coordinates received by a user. The data set $E \equiv \{\gamma_A(\tau^A), \{\tau^A\}\}$ is called the standard data set.

The location problem with respect to $E$, also called the standard location problem or the $E$-location problem for short, is the problem of finding the coordinates $\{x^\alpha\}$ of the user from the sole data $E$ by solving the following algebraic system of four non-linear equations (called the null propagation equations):

$$(x - \gamma_A)^2 = 0, \quad A = 1, 2, 3, 4. \tag{1}$$

The covariant solution to the standard location problem in Minkowski space-time was already discussed [7,8]. Nevertheless, it remains to be formulated in the framework of an arbitrary inertial coordinate system associated with an inertial observer $u$ ($u^2 = -1$), that is, by describing space-time from its relative splitting in space plus time with respect to $u$. Then, the unknown space-time position $x$ of the user can be split, relative to $u$, in inertial components $\{x^0, \vec{x}\}$ as follows:

$$x = x^0 u + \vec{x}, \quad x^0 = -x \cdot u, \quad \vec{x} \cdot u = 0. \tag{2}$$

From now on, we take the speed of light in vacuum to be $c = 1$ and $t \equiv x^0$ (see Appendix A for the notation used in this article).

The matter is then how to determine, with respect to $u$, the solution of the standard location problem, that is, the coordinate transformation from the emission to inertial coordinates $x^0(\tau^A)$ and $\vec{x}(\tau^A)$ when the motions of the emitters are known (received data) in the inertial coordinate system. For this purpose, the tensor quantities that are intrinsically related to the configuration of the emitters at $x$ have to be split in time-like and space-like components. Once the splitting is accomplished, we can recover Kleusberg's analytical solution [3] used in GPS navigation [18].

The covariant solution [7,8] was used in the construction of numerical algorithms for positioning in flat and curved space-times [19–22]. The statement of the location problem in the exact Schwarzschild metric and its perturbative treatment was modelled in [23–25].

The paper is organised as follows. In Section 2, the geometric objects (vector and bivectors) associated with the configuration of the emitters for the reception event are decomposed in time-like and space-like components, relative to an inertial observer. Section 3 is devoted to splitting, relative to an inertial observer, the covariant formula that gives the location of a user in relativistic positioning. In Section 4, we use the preceding splittings to express Kleusberg's procedure in a relativistic formalism and to recover Kleusberg's solution from the covariant solution. In Section 5, an alternative expression of the covariant solution in terms of the principal directions of the configuration bivector is provided. The principal directions are split relative to an inertial observer and Kleusberg's solution is again recovered. The results are summarised and discussed in Section 6. Appendix A is devoted to summarising the notation and conventions used.

Some preliminary results of this work were communicated without proof at the ESA-Advanced Concepts Team Workshop "Relativistic Positioning Systems and their Scientific Applications" (see Ref. [26]).

## 2. Configuration of the Emitters: Underlying Geometry

In relativistic positioning terminology, the  configuration of the emitters for an event $P$ is the set of four events $\{\gamma_A(\tau^A)\}$ of the emitters at the emission times $\{\tau^A\}$ received at $P$. The covariant solution [7] to the standard location problem depends on the configuration of the emitters through different scalars, vectors and bivectors, all of which are computable from the standard data set $\{\gamma_A(\tau^A), \{\tau^A\}\}$. These quantities are analysed in this section.

The set $\mathcal{R}$ of events that are reached by the broadcast signals are called the emission region of the RPS. Then, if $P \in \mathcal{R}$, let us denote by $x \equiv OP$ the position vector with respect the origin $O$ of a given inertial system. If a user at $P$ receives the broadcast times $\{\tau^A\}$ and $\gamma_A$ denotes the position vectors of the emitters at the emission times, then $\gamma_A \equiv O\gamma_A(\tau^A)$. The trajectories followed by the light signals from the emitters $\gamma_A(\tau^A)$ to the reception event $P$ are described by the vectors $m_A \equiv x - \gamma_A$. Let us choose a reference emitter (say $A = 4$) and refer the other emitters (referred emitters) to it. Then, the position vector of the $a$-th emitter with respect to the reference emitter is written as (see Figure 1a)

$$e_a = \gamma_a - \gamma_4 = m_4 - m_a \quad (a = 1, 2, 3). \tag{3}$$

The world function [27,28] of the end point of $e_a$ is the scalar

$$\Omega_a = \frac{1}{2}(e_a)^2. \tag{4}$$

As discussed in  [7], the emission/reception conditions imply $\Omega_a > 0$.

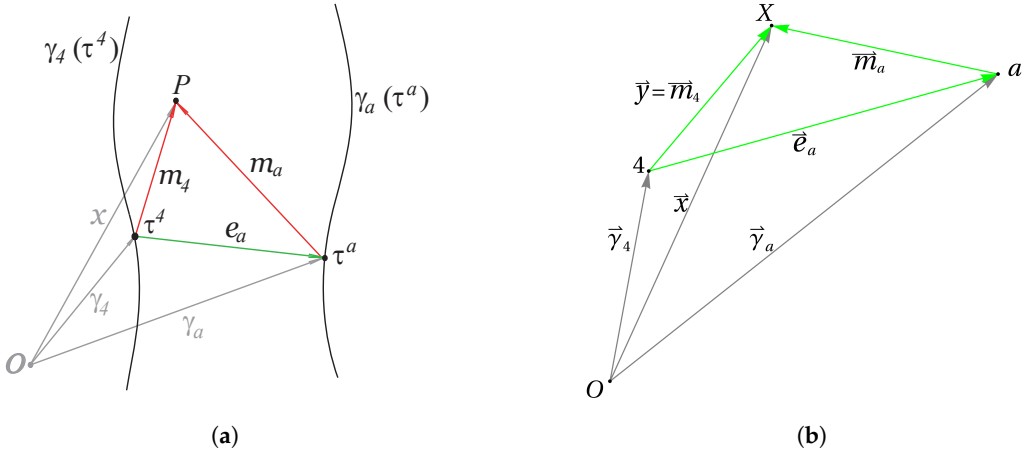

(**a**)

(**b**)

**Figure 1.** Configuration vectors in Minkowski space-time and in the 3-space orthogonal to $u$, $E_\perp$. (**a**) The fourth emitter $\gamma_4$ is taken as the reference emitter. Then, the position vector of the event $P$ is $m_4$. The relative positions $e_a$ of the referred emitters $\gamma_a$ are given by $e_a = \gamma_a - \gamma_4$ ($a = 1, 2, 3$). This figure was taken from [7,8,26]. (**b**) In the 3-space orthogonal to $u$, $E_\perp$, taking the fourth emitter as the reference emitter, $\vec{y} = \vec{m}_4$ is the position vector of the user's location $X$. The relative positions $\vec{e}_a$ of the referred emitters are given by $\vec{e}_a = \vec{\gamma}_a - \vec{\gamma}_4 = \vec{m}_4 - \vec{m}_a$ ($a = 1, 2, 3$).

### 2.1. Splitting of $\gamma_A$, $m_A$ and $e_a$

The position vectors $\gamma_A$ can be decomposed, relative to $u$, as

$$\gamma_A = t_A u + \vec{\gamma}_A \quad (A = 1, 2, 3, 4), \tag{5}$$

where $t_A \equiv \gamma_A^0$ is the coordinate inertial time of the event $\gamma_A(\tau^A)$ as measured by the inertial observer $u$. The vectors $m_A$ are decomposed as follows:

$$m_A = (t - t_A)u + \vec{x} - \vec{\gamma}_A = m_A^0 u + \vec{m}_A \quad (A = 1, 2, 3, 4), \tag{6}$$

which are null and future pointing:

$$(t - t_A)^2 = (\vec{x} - \vec{\gamma}_A)^2, \quad t > t_A. \tag{7}$$

For later convenience, we define

$$y \equiv x - \gamma_4 = m_4 = y^0 u + \vec{y}. \tag{8}$$

The position vector $e_a$ of the $a$-th emitter with respect to the reference emitter is decomposed as follows (see Figure 1b):

$$e_a = \sigma_a u + \vec{e}_a, \quad \sigma_a = t_a - t_4, \quad \vec{e}_a = \vec{\gamma}_a - \vec{\gamma}_4 \quad (a = 1, 2, 3). \tag{9}$$

With the inertial components of $e_a$, $\{\sigma_a, \vec{e}_a\}$, we can express the world function (4) as

$$\Omega_a = \frac{1}{2}((\vec{e}_a)^2 - \sigma_a^2) = \frac{1}{2}[(\vec{\gamma}_a - \vec{\gamma}_4)^2 - (t_a - t_4)^2]. \tag{10}$$

*2.2. Splitting of the Configuration Vector $\chi$*

Here, we always consider that for the reception event $P$, the emitter configuration is regular, that is, the four emission events $\{\gamma_A(\tau^A)\}$ determine a hyperplane named the configuration hyperplane for $P$. Or equivalently, we assume that the configuration vector $\chi$, defined as

$$\chi \equiv *(e_1 \wedge e_2 \wedge e_3) \tag{11}$$

is nonzero, i.e., $\chi \neq 0$. The star $*$ stands for the Hodge dual operator associated with the metric volume element $\eta = (\eta_{\alpha\beta\mu\nu})$. One has $\chi_\alpha = \eta_{\alpha\beta\mu\nu} e_1^\beta e_2^\mu e_3^\nu \neq 0$. Non-regular emitter configurations (with $\chi = 0$) can sporadically occur in current global navigation systems, as it was stressed and considered in [4,5].

Substituting (3) into (11) and taking into account (A3) and (A4), we have

$$\chi = \sigma_1 *(u \wedge \vec{e}_2 \wedge \vec{e}_3) + \sigma_2 *(\vec{e}_1 \wedge u \wedge \vec{e}_3) + \sigma_3 *(\vec{e}_1 \wedge \vec{e}_2 \wedge u) + *(\vec{e}_1 \wedge \vec{e}_2 \wedge \vec{e}_3)$$
$$= (\vec{e}_1, \vec{e}_2, \vec{e}_3) u + \sigma_1 \vec{e}_2 \times \vec{e}_3 + \sigma_2 \vec{e}_3 \times \vec{e}_1 + \sigma_3 \vec{e}_1 \times \vec{e}_2,$$

where $\times$ denotes the cross product between vectors in the three-space orthogonal to $u$, $E_\perp$ (see Appendix A). The following result concerning the decomposition of $\chi$ in time-like and space-like components holds.

**Proposition 1.** *Relative to an inertial observer $u$, the configuration vector is expressed as $\chi = \chi^0 u + \vec{\chi}$, with*

$$\chi^0 = (\vec{e}_1, \vec{e}_2, \vec{e}_3), \qquad \vec{\chi} = \frac{1}{2} \epsilon^{abc} \sigma_a \vec{e}_b \times \vec{e}_c , \tag{12}$$

*where $\{\sigma_a, \vec{e}_a\}$ are the components of the position vectors of the referred emitters.*

Then, we see that $|\chi^0|$ is the volume of the parallelepiped defined by the relative positions $\vec{e}_a$ of the referred emitters. On the other hand, $\vec{\chi}$ represents a weighted vector area. The area $A_{bc}$ of the face generated by $\vec{e}_b$ and $\vec{e}_c$ is weighted with a complementary $\sigma_a$ factor. Then,

$$|\sigma_a \vec{e}_b \times \vec{e}_c| = c|t_c - t_4| A_{bc} \tag{13}$$

is the volume of the time-like parallelepiped generated by $\{\sigma_a u, \vec{e}_b, \vec{e}_c\}$.

*2.3. Splitting of the Bivectors $E^a$*

The positions of the referred emitters, $e_a$, generate the bivectors $E^a$, which are defined as

$$E^1 = *(e_2 \wedge e_3), \quad E^2 = *(e_3 \wedge e_1), \quad E^3 = *(e_1 \wedge e_2), \tag{14}$$

that is, $E^a = *(e_{a+1} \wedge e_{a+2})$, where the notation is understood to be modulo 3. By using index notation, one can write, for example,

$$(e_a \wedge e_b)^{\mu\nu} = e_a^\mu e_b^\nu - e_b^\mu e_a^\nu, \tag{15}$$

$$[*(e_a \wedge e_b)]_{\alpha\beta} = \frac{1}{2}\eta_{\alpha\beta\mu\nu}(e_a \wedge e_b)^{\mu\nu} = \eta_{\alpha\beta\mu\nu}e_a^\mu e_b^\nu. \tag{16}$$

Then, according to (A3), one obtains

$$i(u) * (e_a \wedge e_b) = -\vec{e}_a \times \vec{e}_b \tag{17}$$

where $i()$ is the interior product (defined in Appendix A) and we have taken (9) into account.

For an observer $u$, the electric and magnetic parts of a bivector $E^a$ are vectors in the three-space orthogonal to $u$, which are defined as

$$\vec{S}^a = -i(u)E^a, \qquad \vec{B}^a = -i(u) * E^a, \tag{18}$$

respectively. Then, from (17) and taking into account the identity $*(*E^a) = -E^a$ and (9), we have

$$\vec{S}^a = -i(u) * (e_{a+1} \wedge e_{a+2}) = \vec{e}_{a+1} \times \vec{e}_{a+2}, \tag{19}$$

$$\vec{B}^a = i(u)(e_{a+1} \wedge e_{a+2}) = -\sigma_{a+1}\vec{e}_{a+2} + \sigma_{a+2}\vec{e}_{a+1}, \tag{20}$$

where these equalities are understood to be modulo 3.

Thus, the following result is established.

**Proposition 2.** *Relative to an inertial observer $u$, the configuration two-forms are expressed as*

$$E^a = u \wedge \vec{S}^a - *(u \wedge \vec{B}^a), \tag{21}$$

$$\vec{S}^a \equiv \vec{e}_{a+1} \times \vec{e}_{a+2}, \qquad \vec{B}^a \equiv \epsilon^{abc}\sigma_c \vec{e}_b, \tag{22}$$

*where $\{\sigma_a, \vec{e}_a\}$ are the components of the position vectors of the referred emitters.*

*2.4. Splitting of the Configuration Bivector H*

For each emitter $\gamma_A$, we can define a configuration bivector $H_{(A)}$ with respect to that emitter. In the present work, we define $H \equiv H_{(4)}$ as the configuration bivector with respect to the reference emitter $\gamma_4$:

$$H = \Omega_a E^a = \Omega_1 E^1 + \Omega_2 E^2 + \Omega_3 E^3. \tag{23}$$

Relative to an inertial observer $u$, this bivector can be written as

$$H = u \wedge \vec{S} - *(u \wedge \vec{B}), \tag{24}$$

where $\vec{S} = -i(u)H$ and $\vec{B} = -i(u) * H$ are, respectively, the electric and magnetic parts of $H$ relative to $u$. Taking into account (21)–(23), $\vec{S}$ and $\vec{B}$ are given according to the following proposition.

**Proposition 3.** *Relative to an inertial observer $u$, the electric and magnetic parts, $\vec{S}$ and $\vec{B}$, into which the configuration bivector $H$ is split, can be expressed as*

$$\vec{S} = \Omega_a \vec{S}^a = \Omega_1(\vec{e}_2 \times \vec{e}_3) + \Omega_2(\vec{e}_3 \times \vec{e}_1) + \Omega_3(\vec{e}_1 \times \vec{e}_2), \tag{25}$$

$$\vec{B} = \Omega_a \vec{B}^a = (\Omega_3\sigma_2 - \Omega_2\sigma_3)\vec{e}_1 + (\Omega_1\sigma_3 - \Omega_3\sigma_1)\vec{e}_2 + (\Omega_2\sigma_1 - \Omega_1\sigma_2)\vec{e}_3. \tag{26}$$

As was noted in [8], since the invariant $H_{\mu\nu}(*H)^{\mu\nu}$ identically vanishes, it always occurs that $\vec{S} \cdot \vec{B} = 0$, which also results from (25) and (26).

We can now express the splitting of the configuration vector $\chi$ (12) in terms of $\vec{S}$ and $\vec{B}$, which follows from (25) and (26) by the scalar and cross product with $\vec{e}_a$.

**Proposition 4.** *Relative to an inertial observer u, the configuration vector is expressed as* $\chi = \chi^0 u + \vec{\chi}$, *with*

$$\chi^0 = \frac{\vec{S} \cdot \vec{e}_a}{\Omega_a}, \qquad \vec{\chi} = \frac{\sigma_a \vec{S} + \vec{e}_a \times \vec{B}}{\Omega_a}, \tag{27}$$

*for any* $a = 1, 2, 3$, *where* $\{\sigma_a, \vec{e}_a\}$ *are the components of the position vectors of the referred emitters.*

### 3. The Location Problem

The *E*-location problem in flat space-time was analysed in [7,8] by specifically considering an inertial coordinate system $\{x^\alpha\}$. The result may be expressed in a closed formula that we explain below. The goal of this section is to separate this formula in time-like and space-like components by splitting, relative to an inertial observer, the quantities involved in the covariant solution to the *E*-location problem.

The transformation from emission to inertial coordinates is expressed in closed form according to the following proposition (see [7,8]).

**Proposition 5.** *Let* $\gamma_A(\tau^A)$ *be the world-lines of four arbitrary emitters of an RPS with respect to an inertial coordinate system* $\{x^\alpha\}$, *and let* $\{\tau^A\}$ *be their emission coordinates. The coordinate transformation* $x = K(\tau^A)$ *between the emission and inertial coordinates is given by*

$$x = \gamma_4 + y_* + \lambda \chi, \tag{28}$$

$$y_* = \frac{1}{\xi \cdot \chi} i(\xi) H, \quad \lambda = -\frac{y_*^2}{(y_* \cdot \chi) + \hat{\epsilon}\sqrt{\Delta}}, \quad \Delta = (y_* \cdot \chi)^2 - y_*^2 \chi^2, \tag{29}$$

*where* $\xi$ *is any vector that satisfies the transversality condition* $\xi \cdot \chi \neq 0$; $\hat{\epsilon}$ *is the orientation of the positioning system at* $x$, *which is given by* $\hat{\epsilon} = sgn[*(m_1 \wedge m_2 \wedge m_3 \wedge m_4)]$; $\chi$ *is the configuration vector; and* $H$ *is the configuration bivector.*

As set out in [7], the null propagation Equation (1) can be expressed with respect to the reference emitter and separated into a quadratic equation:

$$y^2 = 0, \tag{30}$$

and a system of three linear equations

$$e_a \cdot y = \Omega_a, \qquad a = 1, 2, 3. \tag{31}$$

The quantity $y_*$ is the particular solution to the system (31). Note that $\xi \cdot y_* = 0$ and that $y_*$ is directly computable from the sole standard emission data since $\chi$ and $H$ are determined by the vectors $e_a$ given by Equation (3).

Furthermore, the consistency of the above definition of $\lambda$ is assured. Since the vectors $\{y_*, \chi\}$ and $\{m_4, \chi\}$ generate the same two-plane, one has that $sgn(\Delta) = sgn[(\chi \cdot m_4)^2]$, and then $\Delta \geqslant 0$.

As was noticed in [8], $\Delta$ is the scalar invariant of $H$ defined by $f \equiv \frac{1}{2} tr H^2$,

$$\Delta = f = -\frac{1}{2} H_{\mu\nu} H^{\mu\nu}, \tag{32}$$

and may be directly computed from $H$.

*3.1. Splitting of the Particular Solution $y_*$*

In this subsection, we carry out the relative decomposition of the particular solution $y_*$ that appears in Equation (28). From Equation (29), the splitting of $y_*$ is obtained by splitting the vector $i(\xi)H$. To begin with, notice that the transversal vector $\xi$ can always be chosen so that its time-like component $\xi^0$ is equal to one ($\xi^0 = -\xi \cdot u = 1$), that is

$$\xi = u + \vec{\xi}. \tag{33}$$

Thus, the transversality condition says that $\chi^0 \neq \vec{\xi} \cdot \vec{\chi}$, and from Proposition 1, this other result follows.

**Proposition 6.** *Relative to an inertial observer $u$, the transversality condition $\xi \cdot \chi \neq 0$ is expressed as*

$$(\vec{e}_1, \vec{e}_2, \vec{e}_3) \neq \frac{1}{2} \epsilon^{abc} \sigma_a \left(\vec{\xi}, \vec{e}_b, \vec{e}_c\right). \tag{34}$$

Then, from (33) and (24), we have

$$i(\xi)H = -\vec{S} - (\vec{\xi} \cdot \vec{S})u + *(\xi \wedge u \wedge \vec{B}) = -(\vec{\xi} \cdot \vec{S})u - \vec{S} - \vec{\xi} \times \vec{B}$$

and the following statement holds.

**Proposition 7.** *Relative to an inertial observer $u$, the particular solution $y_*$ orthogonal to $\xi = u + \vec{\xi}$ is expressed as*

$$y_* = y_*^0 u + \vec{y}_*, \tag{35}$$

*where*

$$y_*^0 = -\frac{\vec{\xi} \cdot \vec{S}}{D}, \qquad \vec{y}_* = -\frac{\vec{S} + \vec{\xi} \times \vec{B}}{D}, \tag{36}$$

*with the transversality condition expressed as*

$$D \equiv \xi \cdot \chi = \vec{\xi} \cdot \vec{\chi} - (\vec{e}_1, \vec{e}_2, \vec{e}_3) \neq 0. \tag{37}$$

The vectors $\vec{\chi}$, $\vec{S}$ and $\vec{B}$ are obtained from the positioning data using Equations (12), (25) and (26).

*3.2. Splitting of the Covariant Solution $x$*

Equation (28) gives the solution $x$ for the *E*-location problem provided that $y_*$ and $\hat{e}$ are obtained from a standard set of data $E \equiv \{\gamma_A(\tau^A), \{\tau^A\}\}$. Relative to an inertial observer $u$, the solution $x$ is split as $x = tu + \vec{x}$, with

$$t = t_4 + y_*^0 + \lambda \chi^0, \quad \vec{x} = \vec{\gamma}_4 + \vec{y}_* + \lambda \vec{\chi}. \tag{38}$$

In fact, according to Equation (28), the determination of the scalar $\lambda$ involves both $y_*$ (given by (29)) and $\hat{e}$. Propositions 1 and 3 provide, respectively, the inertial components of $\chi$ and $H$ in terms of the data. Proposition 7 allows for determining $y_* = \{y_*^0, \vec{y}_*\}$. The invariant $\Delta$ can be computed from (24) and (32) to see that it has a clear geometric meaning according to the following proposition.

**Proposition 8.** *Relative to an inertial observer $u$, $H$ splits into electric ($\vec{S}$) and magnetic ($\vec{B}$) parts so that the invariant $\Delta$ is expressed as*

$$\Delta = \vec{S}^2 - \vec{B}^2. \tag{39}$$

According to this proposition, the user knows when it is crossing the region where $\Delta = 0$ (44), namely, when $\vec{S}$ and $\vec{B}$ are equimodular. In this region, the user can still locate itself using Proposition 5.

Then, we can obtain $\lambda$:

$$\lambda = -\frac{-(y_*^0)^2 + \vec{y}_*^{\,2}}{(-y_*^0 \chi^0 + \vec{y}_* \cdot \vec{\chi}) + \hat{\epsilon}\sqrt{\vec{S}^2 - \vec{B}^2}}. \tag{40}$$

Notice that to make this expression for $\lambda$ operative, one needs to determine the orientation $\hat{\epsilon}$, whose very definition (see Proposition 5) involves the unknown $x$. Therefore, the covariant solution of the standard location problem, given by (28), has to be accompanied by a method for obtaining $\hat{\epsilon}$ that does not involve the previous calculation of $x$.

This question was discussed in [7], which showed that in the central region of the positioning system (the space-time region of all $x \in \mathcal{R}$ such that $\chi^2 \leq 0$, see Section 3.3 below), the orientation $\hat{\epsilon}$ is constant and can be obtained as $\hat{\epsilon} = sgn(u \cdot \chi)$ for any future-pointing time-like vector $u$. In particular, if $u$ is an inertial observer,

$$\hat{\epsilon} = -sgn(\chi^0) = -sgn[(\vec{e}_1, \vec{e}_2, \vec{e}_3)]. \tag{41}$$

Note that from (12) and (27):

$$(\vec{e}_1, \vec{e}_2, \vec{e}_3) = \frac{\vec{S} \cdot \vec{e}_a}{\Omega_a}, \quad \text{for any } a = 1, 2, 3, \tag{42}$$

and therefore, since $\Omega_a > 0$,

$$\hat{\epsilon} = -sgn(\vec{S} \cdot \vec{e}_a). \tag{43}$$

Thus, we arrive at the following result.

**Proposition 9.** *In the central region of an RPS, the orientation $\hat{\epsilon}$ is $+1$ or $-1$ if $\vec{S} \cdot \vec{e}_a$ ($a = 1, 2, 3$) is, respectively, negative or positive.*

The determination of $\hat{\epsilon}$ from a set of observational data, as well as its connection with the solution to the bifurcation problem (outside the central region), were analysed elsewhere [7,8], where it was showed that the applicability of Proposition 5 has no strings attached. A brief account of the bifurcation problem follows.

### 3.3. Emission Coordinate Domains and Bifurcation Problem

Equation (28) is the coordinate transformation from the emission to inertial coordinates, $x^\alpha = K^\alpha(\tau^A)$. The inverse transformation $\Theta$, mapping to every $x$ its emission coordinates, $\tau^A = \Theta^A(x)$, is known as the characteristic emission function. As shown in [8], if $j_\Theta(x)$ is the Jacobian determinant of $\Theta$:

$$j_\Theta(x) = 0 \quad \text{iff} \quad \Delta = 0. \tag{44}$$

This property defines two different emission coordinate systems: the front emission coordinate system and the back emission coordinate system. As long as we remain in space-time regions where light signals do not bifurcate, the way emission coordinates are created by the relativistic positioning system imposes that the coordinate domains in space-time of the front and of the back emission coordinate systems are disjoint.

In addition, these coordinate domains are related by the following property: all of the values of the emission coordinates $\{\tau^A\}$ on the back emission coordinate domain are also values of the emission coordinates on a (generically proper) set of the front emission coordinate domain, which is called the time-like front region; the complementary set (in the front emission coordinate domain) of this time-like front region is called the central region of the relativistic positioning system [7,8].

This coincidence of values of the emission coordinates in the back coordinate domain and in the time-like front coordinate domain is at the origin of the bifurcation problem. How can the users of the relativistic positioning system know in which of the two coordinate

domains of space-time they are? To answer this question the user needs to compute the causal character of the emitter configuration (at $x$): it is said to be space-like, light-like or time-like if $\chi^2 < 0$, $\chi^2 = 0$ or $\chi^2 > 0$, respectively, at $x$. The regions defined by these conditions are respectively denoted as $\mathcal{C}_s$, $\mathcal{C}_l$ and $\mathcal{C}_t$. The central region is $\mathcal{C} = \mathcal{C}_s \cup \mathcal{C}_l$. The time-like front region $\mathcal{C}_t^F$ and the central region form the front emission coordinate domain. The back emission coordinate domain is the time-like back region $\mathcal{C}_t^B$.

Therefore, depending on the causal character of the configuration vector $\chi$, we distinguish three situations (see Figures 3–5 in [8]):

1.  If $\chi$ is time-like, there is only one emission solution $P$; the other ($P'$) is a reception solution. In this case, the sign of $\hat{\epsilon}$ can be determined from the sole standard emission data $\{\gamma_A(\tau^A), \{\tau^A\}\}$ (see Proposition 9).
2.  If $\chi$ is light-like, there is only one valid emission solution (the other solution is degenerate). The sign of $\hat{\epsilon}$ can be determined from $\{\gamma_A(\tau^A), \{\tau^A\}\}$ (see Proposition 9).
3.  If $\chi$ is space-like, there are two valid emission solutions: in order to determine the sign of $\hat{\epsilon}$, additional observational information is necessary (relative positions of emitters on the user's celestial sphere; see [8]).

## 4. Kleusberg's Solution

In a GNSS, the pseudoranges are modelled by considering the emitters' positions at signal emission, the user and emitter clock biases, and different corrections that affect the signal propagation [29,30]. The pseudorange equations are first linearised around an approximate position and then solved by iterative methods. The user's approximate position is usually obtained using analytic (closed-form) solutions of the equations (neglecting emitter clock biases and signal propagation corrections), where the unknowns are the user's coordinates and clock bias with respect to a certain reference frame and reference time. In this section, we see how the relative formulation of the characteristic quantities of an RPS allows for interpreting Kleusberg's analytical solution to the GPS navigation equations [3].

### 4.1. Concepts and Notations

In [3], the starting point is the measured pseudoranges between the satellites and the user. The pseudoranges $p_A$ are modelled as follows (taking the speed of light $c = 1$):

$$p_A = [(x - x_A)^2 + (y - y_A)^2 + (z - z_A)^2]^{\frac{1}{2}} + \delta T, \tag{45}$$

where $\{x_A, y_A, z_A\}$ are the Cartesian coordinates of the $A$-th satellite ($A = 1, 2, 3, 4$) at the time $t_A$ of emission; $\{x, y, z\}$ are those of the user at the time $t$ of reception; and $\delta T$ is the user's clock *bias*, which is defined as the difference between the user's clock time and the so-called GPS time.

Assuming that the bias $\delta T$ is the same for all the satellites, it can be removed by subtracting one reference pseudorange ($p_4$) from the other three: $d_a := p_a - p_4$, $a = 1, 2, 3$. In the covariant solution, this difference can be identified with $\sigma_a := t_a - t_4$:

$$d_a \longleftrightarrow \sigma_a.$$

Kleusberg denoted the Euclidean positions of the three emitters with respect to the reference emitter as $b_a \, \hat{\mathbf{e}}_a$ and the user's position vector from the reference emitter as $s_0 \hat{\mathbf{e}}$, where $\hat{\mathbf{e}}_a$ and $\hat{\mathbf{e}}$ are unit vectors. In the covariant solution, the position four-vector $e_a$ of the $a$-th emitter with respect to the reference emitter is decomposed relative to the inertial observer $u$ as $e_a = \sigma_a u + \vec{e}_a$, and therefore,

$$b_a \longleftrightarrow |\vec{e}_a|.$$

These and other correspondences are summarised in Table 1, including the electric and magnetic parts of the configuration bivector $H$.

**Table 1.** Identifying Kleusberg's notation and concepts with the time-like and space-like components of those of the RPS solution. In the last row, $\mu = 1$ ($\mu = -1$) stands for the emission (reception) solutions.

| **Kleusberg** | | | **RPS** |
|---|---|---|---|
| Pseudorange difference | $d_a$ | $\sigma_a$ | Coordinate time difference |
| Emitter distance to reference emitter | $b_a$ | $|\vec{e}_a|$ | Emitter distance to reference emitter |
| Unit vector from reference emitter to emitter $a$ | $\hat{\mathbf{e}}_a$ | $\dfrac{\vec{e}_a}{|\vec{e}_a|}$ | Unit vector from reference emitter to emitter $a$ |
| User distance to emitter $a$ | $s_a$ | $|\vec{m}_a|$ | User distance to emitter $a$ |
| User distance to reference emitter | $s_0$ | $|\vec{y}|$ | User distance to reference emitter |
| Unit vector from reference emitter to user | $\hat{\mathbf{e}}$ | $\dfrac{\vec{y}}{|\vec{y}|}$ | Unit vector from reference emitter to user |
| Semi-difference | $\frac{1}{2}(b_a^2 - d_a^2)$ | $\Omega_a$ | World function scalar |
| Three-vector | $\vec{G}$ | $\dfrac{\vec{S}}{4\Omega_1\Omega_2\Omega_3}$ | Electric part of configuration bivector $H$ |
| Three-vector | $\vec{H}$ | $\dfrac{\vec{B}}{\mu 4\Omega_1\Omega_2\Omega_3}$ | Magnetic part of configuration bivector $H$ |

With these correspondences, we can explain Kleusberg's procedure in the RPS notation. The starting point is the following equations, which result from (6), (8) and (9) (see Figure 1b):

$$\vec{e}_a = \vec{\gamma}_a - \vec{\gamma}_4 = \vec{m}_4 - \vec{m}_a = \vec{y} - \vec{m}_a, \ a = 1, 2, 3,$$

which implies that

$$\vec{m}_a^2 = (\vec{y} - \vec{e}_a)^2 = \vec{y}^2 + \vec{e}_a^2 - 2\vec{y} \cdot \vec{e}_a. \tag{46}$$

Furthermore, from (8), as $y = m_4$ is light-like:

$$-(y^0)^2 + \vec{y}^2 = 0. \tag{47}$$

This equation is solved by taking $y^0 = \mu|\vec{y}|$, where $\mu$ can take the values $\pm 1$, with $\mu = 1$ ($\mu = -1$) for emission (reception) solutions.

Also, from (3) and (8), we have (see Figure 1a)

$$m_a = m_4 - e_a = y - e_a,$$

and then

$$-(y^0 - \sigma_a)^2 + (\vec{y} - \vec{e}_a)^2 = 0, \quad a = 1, 2, 3. \tag{48}$$

Therefore,

$$\vec{m}_a^2 = (y^0 - \sigma_a)^2 = (\mu|\vec{y}| - \sigma_a)^2, \quad a = 1, 2, 3, \tag{49}$$

where in the last equality, we use $y^0 = \mu|\vec{y}|$.

Expanding (49):

$$\vec{m}_a^2 = \vec{y}^2 + \sigma_a^2 - 2\mu|\vec{y}|\sigma_a, \quad a = 1, 2, 3. \tag{50}$$

Equating (46) and (50), we obtain the equations solved by Kleusberg using the same notation as in the covariant solution:

$$-\mu|\vec{y}|\sigma_a + \vec{y} \cdot \vec{e}_a = \Omega_a, \tag{51}$$

where we have taken into account (4). Then,

$$|\vec{y}| = \frac{\Omega_a}{-\mu\sigma_a + \hat{\mathbf{e}} \cdot \vec{e}_a}, \quad a = 1, 2, 3, \tag{52}$$

where we use Kleusberg's notation $\hat{\mathbf{e}} = \frac{\vec{y}}{|\vec{y}|}$ to simplify the expression. Note that the sign of $\mu$, which gives the emission or reception character of the solution, has to be maintained for further discussion, even if such a distinction is not made in Kleusberg's procedure.

### 4.2. Kleusberg's Procedure to Obtain the Solution

Kleusberg's procedure yields $\vec{y} = |\vec{y}|\hat{\mathbf{e}}$, and therefore, involves two steps: first, one obtains $\hat{\mathbf{e}}$, and then $|\vec{y}|$ by substituting $\hat{\mathbf{e}}$ in (52).

In order to solve the system of Equation (51), Kleusberg equated the right-hand sides of the first and second of the equations in (52), and of the second and third, to obtain the following equations:

$$\frac{\Omega_a}{-\mu\sigma_a + \hat{\mathbf{e}} \cdot \vec{e}_a} = \frac{\Omega_{a+1}}{-\mu\sigma_{a+1} + \hat{\mathbf{e}} \cdot \vec{e}_{a+1}}, \quad a = 1, 2. \tag{53}$$

He rearranged each of these equations to obtain

$$\left[\frac{\vec{e}_1}{\Omega_1} - \frac{\vec{e}_2}{\Omega_2}\right] \cdot \hat{\mathbf{e}} = \mu\left(\frac{\sigma_1}{\Omega_1} - \frac{\sigma_2}{\Omega_2}\right), \tag{54}$$

$$\left[\frac{\vec{e}_2}{\Omega_2} - \frac{\vec{e}_3}{\Omega_3}\right] \cdot \hat{\mathbf{e}} = \mu\left(\frac{\sigma_2}{\Omega_2} - \frac{\sigma_3}{\Omega_3}\right). \tag{55}$$

And rewrote these equations as

$$\vec{F}_1 \cdot \hat{\mathbf{e}} = U_1, \quad \vec{F}_2 \cdot \hat{\mathbf{e}} = U_2, \tag{56}$$

$$\vec{F}_a \equiv \frac{\vec{e}_a}{\Omega_a} - \frac{\vec{e}_{a+1}}{\Omega_{a+1}}, \quad U_a \equiv \mu\left(\frac{\sigma_a}{\Omega_a} - \frac{\sigma_{a+1}}{\Omega_{a+1}}\right), \quad a = 1, 2. \tag{57}$$

To solve them, Kleusberg started with the following identity and expanded the double cross-product:

$$\hat{\mathbf{e}} \times (\vec{F}_1 \times \vec{F}_2) = \mu(U_2\vec{F}_1 - U_1\vec{F}_2), \tag{58}$$

and defined two three-vectors:

$$\vec{G} \equiv \vec{F}_1 \times \vec{F}_2, \quad \vec{H} \equiv \mu\left(U_2\vec{F}_1 - U_1\vec{F}_2\right). \tag{59}$$

As results from (25) and (26), these vectors are directly related to the vectors $\vec{S}$ and $\vec{B}$, in which the configuration bivector $H$ of the covariant method is split with respect to the inertial observer $u$ (see Table 1).

**Proposition 10.** *The electric and magnetic parts, $\vec{S}$ and $\vec{B}$, in which the bivector $H$ is split relative to an inertial observer $u$, can be expressed as*

$$\vec{S} = 4\Omega_1\Omega_2\Omega_3\vec{G}, \quad \vec{B} = \mu 4\Omega_1\Omega_2\Omega_3\vec{H}, \tag{60}$$

*with $\vec{G}$ and $\vec{H}$ given in (59) and where $\mu = 1$ ($\mu = -1$) corresponds to emission (reception) solutions.*

Now, Equation (58) is written as

$$\hat{\mathbf{e}} \times \vec{S} = \mu\vec{B}, \tag{61}$$

or equivalently, multiplying by $\vec{S}$ (cross-product) from the left:

$$\vec{S} \times (\hat{\mathbf{e}} \times \vec{S}) = \mu \vec{S} \times \vec{B}. \tag{62}$$

Expanding the vector triple product on the left-hand side:

$$\vec{S}^2 \hat{\mathbf{e}} - (\vec{S} \cdot \hat{\mathbf{e}}) \vec{S} = \mu \vec{S} \times \vec{B}. \tag{63}$$

Furthermore,

$$\vec{S} \cdot \hat{\mathbf{e}} = |\vec{S}| \cos \phi, \tag{64}$$

where $\phi \in [0, \pi]$ is the angle formed by $\{\vec{S}, \vec{y}\}$. And from (61):

$$|\vec{B}| = |\vec{S}| \sin \phi. \tag{65}$$

Therefore, when $\phi = \frac{\pi}{2}$, $\vec{S}$ and $\vec{B}$ are equimodular. Squaring (64) and (65) and adding gives the following:

$$\left( \vec{S} \cdot \hat{\mathbf{e}} \right)^2 + \vec{B}^2 = \vec{S}^2 \quad \Leftrightarrow \quad \vec{S} \cdot \hat{\mathbf{e}} = \pm \sqrt{\vec{S}^2 - \vec{B}^2}. \tag{66}$$

Note that in this step, an extra solution is introduced and that both solutions have the same emission or reception character. Substituing (66) in (63), we arrive at the following result.

**Proposition 11.** *The unit vector* $\hat{\mathbf{e}} = \frac{\vec{y}}{|\vec{y}|}$*, which gives the direction from the reference emitter to the user's position, can be expressed only in terms of the electric and magnetic parts of the bivector H as follows:*

$$\hat{\mathbf{e}} = |\vec{S}|^{-2} \left[ \mu \vec{S} \times \vec{B} \pm \vec{S} \sqrt{\vec{S}^2 - \vec{B}^2} \right], \tag{67}$$

*where* $\mu = 1$ $(\mu = -1)$ *corresponds to the emission (reception) solutions.*

To obtain $|\vec{y}|$, Equation (67) is substituted into any of the equations in (52).

**Proposition 12.** *Using the same notation as in the covariant solution relative to an inertial observer* $u$*, Kleusberg's solution* $\vec{y}_K$ *is expressed as*

$$\vec{y}_K = |\vec{y}| \, \hat{\mathbf{e}} = \Omega_a \frac{\vec{S} \times \vec{B} \pm \mu \vec{S} \sqrt{\vec{S}^2 - \vec{B}^2}}{-\vec{S}^2 \sigma_a + (\vec{S} \times \vec{B}) \cdot \vec{e}_a \pm \mu \vec{S} \cdot \vec{e}_a \sqrt{\vec{S}^2 - \vec{B}^2}}, \tag{68}$$

*for any* $a = 1, 2, 3$.

*4.3. Recovering Kleusberg's Solution from the Covariant Solution*

We can compare Equation (51) with those solved in the covariant method (Equations (30) and (31)), written with respect to $u$:

$$e_a \cdot y = \Omega_a \Rightarrow -y^0 \sigma_a + \vec{y} \cdot \vec{e}_a = \Omega_a, \quad a = 1, 2, 3, \tag{69}$$

and

$$y^2 = 0 \Rightarrow (y^0)^2 = \vec{y}^2, \tag{70}$$

where we use $y = y^0 u + \vec{y}$ and $e_a = \sigma_a u + \vec{e}_a$.

Comparing Equations (51), (69) and (70), we realise that in contrast to the covariant method, where, first, the linear system (69) and then the quadratic Equation (70) are solved (the initial system of equations is split into a linear system and a single quadratic equation),

Kleusberg directly solves the quadratic equation. Equation (51) is equivalent to (69) but includes the quadratic condition (70), specifically $y^0 = \mu|\vec{y}|$.

To recover Equation (68) from the covariant solution (28), let us first split the covariant solution $y$ relative to the inertial observer $u$ as $y = y^0 u + \vec{y}$:

$$y^0 = y^0_* + \lambda \chi^0, \quad \vec{y} = \vec{y}_* + \lambda \vec{\chi}, \tag{71}$$

where $y^0$ and $\vec{y}_*$ are given by (36), $\lambda$ is given by (40), and $\chi^0$ and $\vec{\chi}$ are given by (12).

Note that the particular solution $y_*$ and, by extension, $\lambda$, depend on the choice of the transversal vector $\xi$. Therefore, we expect to recover (68) with a suitable choice of $\xi$. As long as the transversality condition (34) is satisfied, we can choose $\xi$ of the form $\xi = u + \vec{\xi}$. We start by writing a general expression for $\vec{\xi}$ as

$$\vec{\xi} = a\vec{S} + b\vec{B} + c\vec{S} \times \vec{B}, \tag{72}$$

with $\{a, b, c\}$ as real scalars. Kleusberg's solution (68) is equivalent to the covariant solution with respect to the inertial observer (71) if we take $a = \pm \frac{1}{\sqrt{\Delta}}$ and $c = 0$ in (72):

$$\vec{\xi} = \pm \frac{\vec{S}}{\sqrt{\Delta}} + b\vec{B}, \tag{73}$$

with $b$ as a real scalar. With this choice of $\xi$, it is easy to see from (36) that $y_*$ is light-like and, from (40), $\lambda = 0$ such that $\vec{y} = \vec{y}_*$. Then, the following result is established.

**Proposition 13.** *Setting the transversal vector $\xi = u + \vec{\xi}$, with $\vec{\xi} = \pm \frac{\vec{S}}{\sqrt{\Delta}} + b\vec{B}$ and $b$ as a real scalar, the covariant solution $y$ obtained with this choice of $\xi$ is split relative to the inertial observer as $y = y^0 u + \vec{y}$, where*

$$
\begin{aligned}
y^0 &= \Omega_a \frac{\vec{S}^2}{-\vec{S}^2 \sigma_a + (\vec{S} \times \vec{B}) \cdot \vec{e}_a \pm \vec{S} \cdot \vec{e}_a \sqrt{\vec{S}^2 - \vec{B}^2}}, \\
\vec{y} &= \Omega_a \frac{\vec{S} \times \vec{B} \pm \vec{S} \sqrt{\vec{S}^2 - \vec{B}^2}}{-\vec{S}^2 \sigma_a + (\vec{S} \times \vec{B}) \cdot \vec{e}_a \pm \vec{S} \cdot \vec{e}_a \sqrt{\vec{S}^2 - \vec{B}^2}},
\end{aligned}
\tag{74}
$$

*for any $a = 1, 2, 3$. The space-like component $\vec{y}$ is Kleusberg's solution (68).*

## 5. Covariant Solution in Terms of the Principal Directions of $H$

The covariant solution (28) can also be expressed in terms of the principal directions (eigenvectors) of $H$. In general, a two-form, such as $H$, may be algebraically decomposed in terms of its principal directions through its scalar invariants (see [31,32]). We say that a two-form $F$ is regular if it has at least one non-zero scalar invariant and can therefore be decomposed as

$$F = \alpha \, n \wedge l + \beta * (n \wedge l), \tag{75}$$

with $\{\alpha, \beta\}$ as real scalars and $n$ and $l$ as the principal directions of $F$, which are light-like and satisfy $l \cdot n = -1$. These are the eigenvectors of $F$ with the respective eigenvalues $\alpha$ and $-\alpha$ (they are also the eigenvectors of $*F$ with the respective eigenvalues $-\beta$ and $\beta$).

If $f = -\frac{1}{2} F_{\mu\nu} F^{\mu\nu}$ and $\tilde{f} \equiv \frac{1}{2} F_{\mu\nu} (*F)^{\mu\nu}$ are the usual scalar invariants, the eigenvalues are related to the invariants as follows:

$$\alpha = \frac{1}{\sqrt{2}} \sqrt{\sqrt{f^2 + \tilde{f}^2} + f}, \quad \beta = \frac{1}{\sqrt{2}} \sqrt{\sqrt{f^2 + \tilde{f}^2} - f}. \tag{76}$$

As it was noticed in [8], in the case of the bivector $H$, since the invariant $\tilde{f} = H_{\mu\nu}(*H)^{\mu\nu} = 0$, it follows from (76) that $\beta = 0$. Then, from (75), it can be decomposed as

$$H = \alpha \, n \wedge l, \tag{77}$$

with $\alpha = \sqrt{\Delta}$, which follows from (32) and (76). Note that from (44), there is a region where $\Delta = 0$, and thus, $\alpha = 0$. But this does not mean that $H = 0$, rather that $H$ is singular ($\alpha = \beta = 0$) in that region.

*5.1. Principal Directions of H: Covariant Determination*

According to [31,32], the principal directions of a two-form $H$ can be obtained in covariant form from its minimal polynomial. Since the eigenvalue $\beta = 0$, the minimal polynomial $p(\kappa)$ of $H$ is

$$p(\kappa) = (\kappa + \alpha)(\kappa - \alpha)\kappa. \tag{78}$$

For each of the eigenvalues $\pm\alpha$, we can construct a polynomial $p_\pm(\kappa)$:

$$p_+(\kappa) = (\kappa - \alpha)^{-1}p(\kappa), \quad p_-(\kappa) = (\kappa + \alpha)^{-1}p(\kappa). \tag{79}$$

And define the following projectors:

$$\mathcal{H}_+ \equiv p_+(H) = H^2 + \alpha H, \tag{80}$$

$$\mathcal{H}_- \equiv p_-(H) = H^2 - \alpha H. \tag{81}$$

Since $p(H) = H^3 - \alpha^2 H = 0$, it is then easy to verify that the principal directions $n$ and $l$ are obtained by contracting an arbitrary time-like direction $v$ with these projectors:

$$n = i(v)\mathcal{H}_+, \quad l = i(v)\mathcal{H}_-, \tag{82}$$

and normalising such that $n \cdot l = -1$. Now, we can express the covariant solution of the location problem in terms of $l$ and $n$, which are computable from the standard data $E$ using (80)–(82).

**Proposition 14.** *In regions where $\Delta \neq 0$, each of the two solutions $y_+$ and $y_-$ included in the general solution $y$ to the location problem (28) can also be expressed as*

$$y_+ = \frac{\alpha n}{\chi \cdot n} \quad , \quad y_- = -\frac{\alpha l}{\chi \cdot l} \quad , \tag{83}$$

*where n and l are the principal directions of H, which are known from (82), $\alpha = \sqrt{\frac{1}{2}\mathrm{tr}H^2}$ and the configuration vector $\chi$ (11).*

In order to analyse the cases in which the denominators in (83) vanish, first note that we can form a basis of Minkowski space-time $\{l, n, p, q\}$ with the principal directions $l$ and $n$ and with two orthogonal space-like directions $p$ and $q$, where

$$l^2 = n^2 = 0, \ l \cdot n = -1, \ p^2 = q^2 = 1,$$
$$l \cdot p = l \cdot q = n \cdot p = n \cdot q = p \cdot q = 0.$$

We can express the configuration vector $\chi$ in this basis as

$$\chi = a\,l + b\,n + c\,p + d\,q, \tag{84}$$

with $\{a, b, c, d\}$ as real scalars. From (11) and (23), it follows that

$$i(\chi) * H = 0. \tag{85}$$

Since $H = \alpha n \wedge l$ ($\beta = 0$), it follows that $*H = h p \wedge q$ (with $h$ as a real scalar), and thus,

$$i(\chi) * H = h\, i(\chi)(p \otimes q - q \otimes p) = h(\chi \cdot p)q - h(\chi \cdot q)p = 0.$$

Therefore, $c = \chi \cdot p = 0$ and $d = \chi \cdot q = 0$, and then $\chi$ is a linear combination of $l$ and $n$:

$$\chi = a\, l + b\, n. \tag{86}$$

*5.2. Emission Coordinate Domains and Bifurcation Problem*

With this expression of $\chi$, we can analyse the denominators in (83). This discussion is directly related to the causal character of $\chi$, which, in turn, determines the number of valid emission solutions (see Section 3).

1.  If $\chi$ is time-like, there is only one emission solution, while the other is a reception solution. Explicitly using (86),

$$\chi^2 = (a\, l + b\, n)^2 = -2ab < 0. \tag{87}$$

Therefore, $sgn(a) = sgn(b) \neq 0$. Since $a = -\chi \cdot n$ and $b = -\chi \cdot l$, it follows that the denominators in (83) do not vanish in this case. Furthermore, since $n$ and $l$ are both future oriented and $\alpha > 0$, $y_+$ and $y_-$ have different orientations, with one being an emission and the other a reception solution. If $\chi$ is future (past) oriented, then $y_-$ ($y_+$) is the valid emission solution.

2.  If $\chi$ is light-like, there is only one valid emission solution (the other solution is degenerate). Again, using (86),

$$\chi^2 = (a\, l + b\, n)^2 = -2ab = 0. \tag{88}$$

Therefore, $a = 0$ or $b = 0$ such that $\chi$ is collinear with $n$ or $l$ and one of the solutions, $y_+$ or $y_-$, is degenerate, with the other being an emission solution. If $\chi$ is future (past) oriented, then $y_-$ ($y_+$) is the valid emission solution.

3.  If $\chi$ is space-like, there are two valid solutions. Using (86),

$$\chi^2 = (al + bn)^2 = -2ab > 0. \tag{89}$$

Therefore, $sgn(a) = -sgn(b) \neq 0$ and the denominators in (83) do not vanish in this case. Since $n$ and $l$ are both future oriented and $\alpha > 0$, $y_+$ and $y_-$ are both emission solutions if $\chi \cdot n > 0$ or $\chi \cdot l < 0$.

*5.3. User Location in the Region Where $j_\Theta(x) = 0$*

As said earlier, there is a region in which $\alpha = 0$, and then, since $\beta = 0$, $H$ is a singular two-form. In this case, $H$ can be decomposed as

$$H = k \wedge p, \tag{90}$$

where $k$ is the fundamental direction, which is light-like and satisfies $i(k)H = i(k) * H = 0$, and $p$ is a space-like vector such that $k \cdot p = 0$. Note that $p$ can be determined up to a transformation $p \to p + \omega k$ (with $\omega$ as a real scalar). To obtain the fundamental direction $k$, one constructs the projector $\mathcal{H}_0$ by setting $\alpha = 0$ in (80) or (81):

$$\mathcal{H}_0 = H^2 \tag{91}$$

and contracts an arbitrary time-like direction $v$ with it:

$$k = i(v)\mathcal{H}_0. \tag{92}$$

In this case, the only solution to the location problem is found according to the following result.

**Proposition 15.** *In the region where $\Delta = 0$, the solution $y_0$ to the location problem (28) can also be expressed as*

$$y_0 = \Omega_a \frac{k}{k \cdot e_a}, \quad (a = 1, 2, 3),$$ (93)

*where $k$ is the fundamental direction of $H$, which is known from (92), $e_a$ is the position vector of the a-th emitter with respect to the reference emitter (3) and $\Omega_a$ is the world function scalar (4).*

Note that from (11) and (23), $i(e_a)H = -\Omega_a\chi$. Furthermore, as was stated in [8], the region where $\Delta = 0$ is a subregion of the time-like region $\mathcal{C}_t$ such that $\chi^2 > 0$ in this region. Therefore, $(i(e_a)H)^2 \neq 0$. Explicitly, using (90):

$$i(e_a)H = -\Omega_a[(k \cdot e_a)p - (p \cdot e_a)k].$$ (94)

Squaring:

$$(i(e_a)H)^2 = \Omega_a^2(k \cdot e_a)^2 \neq 0.$$ (95)

Hence, the denominator in (93) does not vanish in this region.

*5.4. Splitting of the Covariant Solution in Terms of l and n*

We can split the principal directions of $H$ relative to an inertial observer $u$ by posing the eigenvalue equations

$$i(n)H = \alpha n, \qquad i(l)H = -\alpha l,$$ (96)

and expressing $H$ relative to $u$ according to (24) and $n$ and $l$ as

$$n = n^0 u + \vec{n}, \qquad l = l^0 u + \vec{l}.$$ (97)

Expanding the left-hand side of (96) using (24), (97) and (A3):

$$i(n)H = i(n)(u \wedge \vec{S}) - i(n) * (u \wedge \vec{B}) = (-\vec{n} \cdot \vec{S})u - (n^0\vec{S} + \vec{n} \times \vec{B}).$$ (98)

Expanding the right-hand side of (96) using (97):

$$i(n)H = \alpha n^0 u + \alpha\vec{n},$$ (99)

and Equations (98) and (99), yields the following equations for the time-like and space-like components of the principal direction $n$:

$$\vec{n} \cdot \vec{S} = -\alpha n^0, \qquad n^0\vec{S} + \vec{n} \times \vec{B} = -\alpha\vec{n}.$$ (100)

Similarly, the second eigenvalue equation in (96) leads to the following equations for the time-like and space-like components of the principal direction $l$:

$$\vec{l} \cdot \vec{S} = \alpha l^0, \qquad l^0\vec{S} + \vec{l} \times \vec{B} = \alpha\vec{l}.$$ (101)

The solution of Equations (100) and (101) yields the following result.

**Proposition 16.** *In each coordinate domain of an RPS, the bivector $H$ is a regular two-form that can be algebraically decomposed as $H = \alpha\, n \wedge l$, where $n$ and $l$ are the prinicipal directions, which*

*are light-like eigenvectors of H with eigenvalues α and −α and satisfy n · l = −1. Relative to an inertial observer u, the principal directions are split as $n = n^0 u + \vec{n}$ and $l = l^0 u + \vec{l}$, where*

$$n^0 = \frac{\vec{S}^2}{M}, \qquad \vec{n} = \frac{-\alpha\vec{S} + \vec{S} \times \vec{B}}{M}, \tag{102}$$

$$l^0 = \frac{\vec{S}^2}{M}, \qquad \vec{l} = \frac{\alpha\vec{S} + \vec{S} \times \vec{B}}{M}, \tag{103}$$

*with $M = \sqrt{2}\alpha|\vec{S}|$ and $\alpha = \sqrt{\vec{S}^2 - \vec{B}^2}$.*

Then, the alternative expression of the covariant solution given in Proposition 14 may be split according to the following proposition.

**Proposition 17.** *Relatively to an inertial observer u, the covariant solutions $y_+$ and $y_-$ for the E-location problem given in (83) are split as $y_+ = y_+^0 u + \vec{y}_+$ and $y_- = y_-^0 u + \vec{y}_-$, with*

$$y_+^0 = Nn^0, \qquad \vec{y}_+ = N\vec{n}, \qquad N = \frac{\alpha}{\chi \cdot n}, \tag{104}$$

$$y_-^0 = -Ll^0, \qquad \vec{y}_- = -L\vec{l}, \qquad L = \frac{\alpha}{\chi \cdot l}, \tag{105}$$

*where $\{n^0, \vec{n}\}$ and $\{l^0, \vec{l}\}$ are given by (102) and (103), respectively.*

Using Proposition 17, it is easy to verify that each of the solutions included in (68), depending on the sign of the square root term, are $\vec{y}_+$ and $\vec{y}_-$ given in (104)–(105).

**Proposition 18.** *The covariant solutions $y_\pm$ are split relative to the inertial observer as $y_\pm = y_\pm^0 u + \vec{y}_\pm$, where $y_\pm^0$ and $\vec{y}_\pm$ are given by (104)–(105). The space-like component $\vec{y}_\pm$ is Kleusberg's solution (68).*

## 6. Discussion and Comments

Users of global navigation satellite systems, such as GPS and Galileo, locate themselves by solving the navigation equations through iterative methods around an initial approximate position. This initial estimation is usually analytically obtained by solving a simplified version of the equations (neglecting gravitational, atmospheric and instrumental effects). These closed-form solutions are either based on pseudoranges or on pseudorange differences. In [14], a known analytical solution based on pseudoranges, Bancroft's solution [1], was interpreted in the language of RPS.

In this paper we analyse, from the perspective of the theory of RPS, another known closed-form solution based on pseudorange differences, i.e., Kleusberg's solution [3]. We first formulated the theory of RPS in the framework of an inertial coordinate system. To this end, the quantities that are related to the configuration of the emitters, such as the configuration vector $\chi$ and the bivector $H$, are split in time-like and space-like components. This formulation of the theory allowed us to interpret and express Kleusberg's solution to the pseudorange navigation equations in terms of these quantities.

Furthermore, a new expression of the covariant solution to the standard location problem is given in terms of the principal directions (eigenvectors) of $H$ and its only non-vanishing scalar invariant, and the procedure to obtain these eigenvectors in covariant form [31,32] is applied. A brief analysis of the solutions based on the causal character of the emitter configuration is provided for this alternative expression of the covariant solution.

Kleusberg's solution is recovered from the space-like component of the covariant solution, both from its original expression and from the alternative expression given in terms of the principal directions of the bivector $H$.

This manuscript aims to contribute to the development of RPS theory in a specific direction: the $3 + 1$ splitting, relative to an inertial observer, of the general transformation

from emission to inertial coordinates in flat space-time. But RPS theory [11] was conceived as a primordial element to provide current GNSSs with a true relativistic conception that was founded exclusively on relativity theory. Moreover, since an RPS can be constructed without any information about the gravitational field [33], it allows for carrying out relativistic gravimetry [34,35] in the unknown space-time domain where it operates, which is an essential element of a true relativistic laboratory (see Refs. [9,10] for a deep discussion of these ideas).

**Author Contributions:** All authors contributed equally to prepare the original draft of the manuscript and have closely collaborated to obtain and present the results. All authors have read and agreed to the published version of this manuscript.

**Funding:** We would like to thank the support from the Spanish Ministerio de Ciencia, Innovación y Universidades, Projects PID2019-109753GB-C21/AEI/10.13039/501100011033 and PID2019-109753GB-C22/AEI/10.13039/501100011033, and from the Conselleria d'Educació, Universitats i Ocupació, Generalitat Valenciana, Project CIAICO/2022/252.

**Data Availability Statement:** No new data were created or analysed in this study. Data sharing is not applicable to this article.

**Acknowledgments:** J.J. Ferrando and J.A. Morales-Lladosa would like to thank Bartolomé Coll for introducing us to the theory of RPS, among other topics, and encouraging us to develop the theory.

**Conflicts of Interest:** The authors declare no conflicts of interest.

## Appendix A. Notation

The main sign convections and notations we adopt in this paper are as follows:

(i) $g$ is the Minkowski space-time metric, with the signature taken as $(-,+,+,+)$. We use units in which the speed of light in vacuum is $c = 1$.

(ii) $\eta$ is the metric volume element of $g$, as defined by $\eta_{\alpha\beta\gamma\delta} = -\sqrt{-\det g}\,\epsilon_{\alpha\beta\gamma\delta}$, where $\epsilon_{\alpha\beta\gamma\delta}$ stands for the Levi–Civita permutation symbol $\epsilon_{0123} = 1$. The Hodge dual operator associated with $\eta$ is denoted by an asterisk $*$.

For instance, and using index notation, if $x$, $y$ and $z$ are space-time vectors, one has

$$[*(x \wedge y \wedge z)]_\alpha = \eta_{\alpha\beta\gamma\delta}x^\beta y^\gamma z^\delta. \tag{A1}$$

where $\wedge$ stands for the *wedge or exterior product* (antisymmetrised tensorial product of antisymmetric tensors).

(iii) $i()$ denotes the interior or contracted product, that is, $i(x)T$ denotes the contraction of a vector $x$ and the first slot of a tensor $T$. Thus, for a covariant two-tensor, $[i(x)T]_\nu = x^\mu T_{\mu\nu}$.

(iv) For an observer of unit velocity $u$, $(u^2 = g(u,u) = -1)$, the vector $x$ splits as

$$x = x^0 u + x_\perp \tag{A2}$$

where $x^0 = -g(x,u) \equiv -x \cdot u$ and $x_\perp = \vec{x}$ is orthogonal to $u$, $g(u,x_\perp) = 0$. We use the notation $x = (x^0, \vec{x})$, where $x^0$ and $x_\perp (\in E_\perp)$ are the time-like and space-like components $x$ relative to $u$, respectively, and $E_\perp$ is the space orthogonal to $u$. $E_\perp$ has an induced volume element given by $\eta_\perp \equiv -i(u)\eta$, that is, $(\eta_\perp)_{\beta\gamma\delta} = -u^\alpha \eta_{\alpha\beta\gamma\delta}$.

(v) For vectors $\vec{x}, \vec{y} \in E_\perp$, the vector or cross-product is expressed as

$$\vec{x} \times \vec{y} = *(u \wedge \vec{x} \wedge \vec{y}), \tag{A3}$$

and, if $\vec{z} \in E_\perp$, the scalar triple product $(\vec{x} \times \vec{y}) \cdot \vec{z} \equiv (\vec{x}, \vec{y}, \vec{z})$ is then given by

$$(\vec{x}, \vec{y}, \vec{z})\,u = *(\vec{x} \wedge \vec{y} \wedge \vec{z}). \tag{A4}$$

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
