# Peer review of "Location Problem in Relativistic Positioning: Relative Formulation"

_universe, doi:10.3390/universe10070299_

Round 1
Reviewer 1 Report
Comments and Suggestions for Authors
The authors consider a 3+1 decomposition of the flat spacetime location problem in relativistic positioning systems (RPSs). They use their formalism to recover and interpret the Kleusberg solution for GPS in their framework. They also present a new expression for the solution to the flat spacetime location problem in terms of the principal direction of a two-form.
This is nice work. One obstacle to the adoption of fundamentally relativistic methods in GNSS is that the relativistic formalism is somewhat removed from those of the standard formulation in terms of pseudoranges, and I think that it is important for the RPS community to relate the relativistic formalism to the navigation equations employed in existing GNSS systems. The calculations in the article appear to be valid (though I must warn that I have not checked them in any detail), and for the most part appear to be a straightforward extension of work done in refs. [7,8]. In short, I think the work is both important and valid, and I think it can be published in Universe.
I only have one optional recommendation for readability: at the end of Eq. (45), it is not obvious to me that the clock bias dT corresponds to a differential or one-form. If the authors mean to indicate that dT is a small quantity, perhaps they can use \delta T instead.
Author Response
Thank you for your comments and for your accurate explanation of the article's contents.
With regard to your last comment, we agree that "\delta T" is more appropriate to indicate small quantities of the user clock bias and to avoid confusion with a one-form.
Reviewer 2 Report
Comments and Suggestions for Authors
In my opinion this is a fine and well written research work.
I could not find any printing errors and the language is fine.
The article is organized such that 18 propositions are proved.
This works well.
It is very seldom that I have no suggestions for improvements of a paper.
But that is the case for this paper.
My conclusion is that the submitted paper can be published witout any changes.
Author Response
Thank you for reading the preprint and for your encouraging comments.